# Stability assessment of roadbed affected by ground subsidence adjacent to urban railways

**Ki-Young Eum[1], Young-Kon Park[2], Sang-Soo Jeon[3]**

[1]Advanced infrastructure research team, Korea Railroad Research Institute, Chuldobakmulgwan-Ro 176, Uiwang City, Gyeonggi-Do 16105, South Korea

[2]Smart station research team, Korea Railroad Research Institute, Chuldobakmulgwan-Ro 176, Uiwang City, Gyeonggi-Do 16105, South Korea

[3]Department of Civil and Urban Engineering, Inje University, Inje-Ro 197, Gimhae City, Kyungsangnam-Do 50834, South Korea

*Correspondence to:* Sang-Soo Jeon (ssj@inje.ac.kr)

**Abstract**. In recent years, leakages in aged pipelines for water and sewage in urban areas have frequently induced ground loss resulting in cavities and ground subsidence causes the roadbed settlement greater than the allowable value. In this study, FLAC$^{3D}$, which is a three-dimensional finite-difference numerical modeling software, is used to do stability and risk level assessment for the roadbed adjacent to urban railways with respect to various groundwater levels and the geometric characteristics of cavities. Numerical results show that roadbed settlement increases as the diameter (D) of the cavity increases and the distance (d) between the roadbed and the cavity decreases. The regression analyses results show that, as D/d is greater than 0.2 and less than 0.3, the roadbed is in the status of caution or warning. It requires a database of measurement sensors for real-time monitoring of the roadbed, structures and groundwater to prevent disasters in advance. As D/d exceeds 0.35, the roadbed settlement, which substantially increases and the roadbed is in the status of danger. Since it may result in highly probable traffic accident, train operation should be stopped and the roadbed should be reinforced or repaired. The effects of groundwater level on the roadbed settlement are examined and the analyses results indicate that a roadbed settlement is highly influenced by groundwater levels to an extent greater than even the influence of the size of the cavity.

## 1   Introduction

Urban railways in South Korea have been initiated from the Seoul subway 1$^{st}$ line in1974 and have been operating in Seoul city and several metropolitan cities. The number of passengers using urban railway are being increased and it has played a significantly important role in public transportation for urban development. Urban railway is defined as transportation facility and method for smooth transportation in the city and includes light rail transit and subway as indicated in the law of urban railway (Ministry of land, 2017).

Risk management associated with safety is a fundamental focus in railway operations. It has been integrated into global safety management system of railways (Berrado et al., 2010) and developed to allow a rapid risk assessment using a common risk score matrix (Braband, 2011). As roadbed settlements exceed the allowable limits, it may result in track irregularity and derailments of trains causing heavy loss of life. Therefore, risk management tools are developed to deal with track safety by controlling and reducing the risk of derailments (Zarembski et al., 2006). In this study, methods to secure the stability of roadbeds have been examined using numerical analysis.

Numerical analyses have been widely used for risk assessment. Numerical analyses using three-dimensional geotechnical codes were carried out to predict the subsidence area and its interaction with buildings (Castellanza et al., 2015) and a three-dimensional groundwater flow model for risk evaluation was developed to be an effective management strategy (Ashfaque et al., 2017). The coupling of numerical models and monitoring data contribute to undertake efficient risk reduction policies (Bozzano et al., 2013). Especially using FLAC, which is a finite-difference numerical code especially specialized in the area of geotechnical engineering, numerical computations to simulate the influence of rainfall (Pisani, 2010), both acoustic emission (AE) activities at AE sensor locations of

the Kannagawa cavern (Cai et al., 2007), and a comprehensive pump test at Sellafield (Hakami, 2001) showed
good agreement with field monitoring results. In this study, FLAC$^{3D}$, which is a three-dimensional finite-difference
numerical code especially specialized in the area of geotechnical engineering, is adopted for numerical analysis.
Research on stability assessment and reinforcement of railway roadbeds has been actively carried out, but the
effect of the cavity adjacent to urban railways on roadbed behavior has rarely studied. In recent years, the number
of accidents induced by cavities larger than 2 m in diameter has increased especially in highly populated cities in
South Korea. Therefore, the residents in these cities were terrified of cavities after the accidents (Shin and Roh,
2006). Especially, ground subsidence near subways due to self-weight and/or surcharge loading was around 60%
(Lee and Kang, 2014). Changes in groundwater levels may cause increased occurrences of ground subsidence
because the lowering of groundwater levels lead to ground settlement (Lee et al., 2015). Groundwater level
influences both ground settlement and stability of underground structures. Deep excavation of the ground adjacent
to urban railways has a significance influence on the allowable tensile strength of underground structures (Lee at
al., 2017). If large underground cavities are located at nearby roadbeds, there is a high potential of ground
subsidence.
Ground subsidence (Fig. 1) in South Korea occurred at nearby urban railways most recently (Kyunghang times,
2016). The ground subsidence (Fig. 1a) occurred with a cavity of depth 5 m, width 8 m, and length 80 m near the
Seokchon subway station in Seoul City. The accident was induced by the inappropriate deep excavation near the
subway. The ground subsidence (Fig. 1b) was caused by the leakage of a water pipeline with a large-scale cavity
of depth 21 m, width 11 m, and length 12 m near Bakchon subway station in Incheon City (Newshankuk, 2016).
The ground subsidence (Fig. 1c) occurred near Samseongjungang subway station. Six cavities were found almost
simultaneously in Seoul City (Kyunghang times, 2016). A small-scale cavity of depth 2.2 m (Fig. 1d) occurred
near Janghanpyeong subway station in Seoul City, but the cause of this accident has not been clarified. The
accident was assumed to be caused by inappropriate recovery construction near subway extension.
Ground subsidence with a cavity having depth 3.6 m (Fig. 1e) occurred as the replacement work of a sewage
pipeline was carried out at Texas in the US (Wikitree, 2016). Ground subsidence with a cavity having a width of
15 m (Fig. 1f) occurred as tunnel excavation work for subway extension was carried out at Fukuoka in Japan
(Chosun Ilbo, 2016). Ground subsidence with a cavity of width 25 m (Fig. 1g) occurred as a 50 m tunnel
excavation near the light rail transit was carried out at Ottawa in Canada (Yonhap news, 2016). Ground subsidence
with a cavity of depth 10 m (Fig. 2h) occurred as subway construction was carried out near Guangzhou in China
(Sisa china, 2016). Ground subsidence with a large-scale cavity in urban areas is highly correlated with the
undiscerned development of urban areas, abuse of groundwater and inappropriate underground construction.

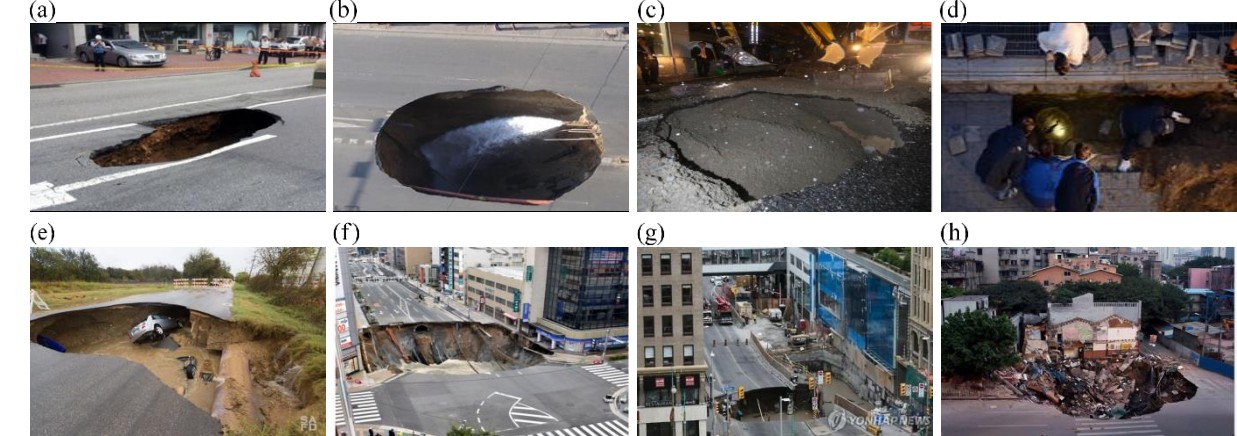

**Figure 1.** Ground subsidence nearby subway of urban railway: **(a)** Seokchon subway station, **(b)** Bakchon subway station, **(c)**
Samseongjungang subway station, **(d)** Janghanpyeong subway station in South Korea, **(e)** Texas in the US, **(f)** Fukuoka in
Japan, **(g)** Ottawa in Canada, and **(h)** Guangzhou in China.

80% of the ground subsidence occurred from 2010 until the beginning of 2014 in Seoul City was induced by
aged pipelines for water and sewage (Oh et al., 2015). Since 48% and 30% of sewage pipelines in Seoul city were
constructed more than thirty and fifty years ago, respectively. Aged pipelines for water and sewage pipelines cause
numerous cavities in the near future (The Segye times, 2016).
As a cavity exists at the center of the railway track in the box structures of urban railways, its influence on box
structures and roadbed settlements has been examined to observe the effects of cavities adjacent to the roadbeds of
urban railways (Lee at al., 2015). A method to establish a database was proposed to prevent and manage the
disasters (Choi et al., 2007).
As a cavity exists adjacent to the roadbed, in this study, a three-dimensional numerical analysis using FLAC[3D] is
carried out to assess both roadbed stability and risk level with respect to the distance between the center of the
roadbed and the center of the cavity, diameter of the cavity, and groundwater levels.

## 2   Numerical analysis

In the following sections, the FLAC[3D] given in this work are briefly described in the following sections by
paraphrasing from those of Itasca Consulting Group (2002).

### 2.1   Theoretical background of FLAC[3D]

FLAC[3D] (Fast Lagrangian Analysis of Continua in three Dimensions) is numerical modeling software for advanced
geotechnical analysis of soil, rock, groundwater, and ground support in three dimensions. FLAC is used for
analysis, testing, and design by geotechnical, civil, and mining engineers (Itasca Consulting Group Inc., 2002). It is
designed to accommodate any kind of geotechnical engineering project that requires continuum analysis. The
mechanics of the medium are derived from general principles (definition of strain, laws of motion), and the use of
constitutive equations defining the idealized material. The resulting mathematical expression is a set of partial
differential equations, relating mechanical (stress) and kinematic (strain rate, velocity) variables, which are to be
solved for particular geometries and properties, given specific boundary and initial conditions. An important aspect
of the model is the inclusion of the equations of motion, although FLAC3D is primarily concerned with the state of
stress and deformation of the medium near the state of equilibrium. Application of the continuum form of the
momentum principle yields Cauchy's equations of motion:

$$\sigma_{ij,j} + \rho b_i = \rho(d_{vi}/d_t) \tag{1}$$

Where $\sigma$ is the symmetric stress tensor, $\rho$ is the mass per unit volume of the medium, $[b]$ is the body force per
unit mass, and $d[v]/dt$ is the material derivative of the velocity. These laws govern, in the mathematical model,
the motion of an elementary volume of the medium from the forces applied to it. Note that in the case of static
equilibrium of the medium, the acceleration $d[v]/dt$ is zero, and Eq. (1) reduces to the partial differential
equations of equilibrium:

$$\sigma_{ij,j} + \rho b_i = 0 \tag{2}$$

### 2.2   Conditions for numerical analysis

The Mohr-Coulomb failure model has been used for the analysis (Itasca Consulting Group Inc., 2015). Since there
are various causes and sizes of the cavity of ground subsidence occurring near urban railway, it is very difficult to
simulate the process of cavity generation. A circular cavity below the ground surface has been modeled with
respect to diameters (D) of 4 -10 m, which is selected by historical events as described in previous section.
Distances of 15-25 m from the cavity to the center of the roadbed and various groundwater levels are arbitrarily
selected for roadbed settlement influenced by given size of cavity. The analysis is performed based on the
configuration of the analysis (Fig. 2). As shown in the figure, roller supports prevent normal translations, but
capable of tangential translations and/or rotations. There is a single linear reaction force in either vertical or
horizontal directions.

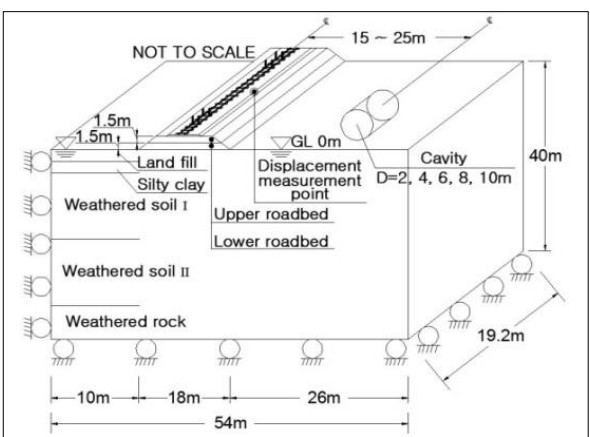

**Figure 2.** Configuration of the railway roadbed and cavity

An embankment consists of the lower roadbed, upper roadbed, and gravel ballast. The roadbed width at the bottom of the ballast is 8m. The widths of its bottom and top are 5.1 and 3.3 m, respectively, and its slope is 1:1.8. In-situ soil consists of reclaimed soil, silty clay, weathered soil, and weathered rock. Its physical properties listed in Table 1 are obtained from lab experiments of soil sampled at a construction site.

KS60 rail and prestressed concrete (PC) sleeper commonly used in gravel ballast have been used for the numerical analysis. A rail pad, which is widely used to minimize vibration and impact loading during train operation is made of ethylene vinyl acetate (EVA). However, in this study, a thermoplastic polyurethane (TPU) rail pad, which is more economical and has higher tensile strength has been used for the numerical analysis. Its properties are listed in Table 1. The beam element is used for the rail and rail pad.

An axial load of the urban railway train (16 tons) is applied for the numerical analysis. The effective loading is estimated by multiplying 1.2 with half of the axial load considering a wheel loading increment of 20% and a marginal safety of deficiency of the cant. Dynamic loading to reflect dynamic impact ratio (Fig. 3) was estimated by multiplying 1.2 with the effective loading (Ministry of land, 2013).

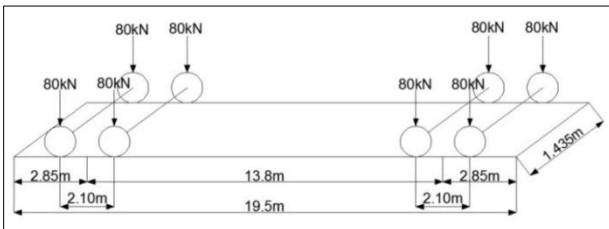

**Figure 3.** Configuration of the train load

In general, an allowable settlement of 10 mm has been recommended in South Korea. The vibratory loading induces the gravel to be in a loose state, and frequent repairs of ballasts are required. Therefore, an allowable settlement of 2.5 mm is used to attain additional marginal safety considering the compressive displacement of both the rail pad and ballasts, settlement of rail, ride quality, and both water inflow and cracks in the pavement surface of roadbeds (Jeon, 2014).

**Table 1.** Physical properties of soil, rail, PC sleeper and the rail pad

| | Soil type | Height (m) | Unit weight (kN/m³) | Elastic modulus (kPa) | Poisson's ratio (υ) | Cohesion (kPa) | Friction angle (°) | Coefficient of permeability (cm/s) | $K_o$ |
|---|---|---|---|---|---|---|---|---|---|
| Soil | Ballast stone | 0.3 | 19.0 | 133,900 | 0.30 | - | 35 | - | 0.43 |
| | Upper roadbed | 1.5 | 18.0 | 81,600 | 0.20 | 3.0 | 32 | - | 0.47 |
| | Lower roadbed | 1.5 | 18.0 | 51,000 | 0.30 | 10.0 | 30 | - | 0.50 |
| | Land fill | 1.5 | 17.0 | 30,000 | 0.35 | 5.0 | 24 | $1.0 \times 10^{-3}$ | 0.59 |
| | Silty clay | 1.5 | 17.0 | 20,000 | 0.35 | 5.0 | 25 | $5.0 \times 10^{-4}$ | 0.58 |
| | Weathered soil Ⅰ | 15.0 | 19.0 | 75,000 | 0.33 | 10.0 | 30 | $1.0 \times 10^{-4}$ | 0.50 |
| | Weathered soil Ⅱ | 15.0 | 19.0 | 70,000 | 0.33 | 10.0 | 33 | $1.0 \times 10^{-4}$ | 0.46 |
| | Weathered rock | 7.0 | 20.0 | 110,000 | 0.31 | 60.0 | 42 | $1.0 \times 10^{-5}$ | 0.33 |

| | Area (mm²) | Unit weight (kN/m³) | Elastic modulus (kPa) | Moment of inertia(m⁴) | |
|---|---|---|---|---|---|
| KS60 rail | | | | $I_{XX}$ | $I_{YY}$ |
| | 7,741 | 77.5 | $21,000 \times 10^4$ | $30,820 \times 10^{-9}$ | $5,120 \times 10^{-9}$ |

| | Length (m) | Width (m) | Height (m) | Interval between sleepers (m) |
|---|---|---|---|---|
| PC sleeper | | | | |
| | 2.45 | 0.28 | 0.20 | 0.58 |

| | Thickness (mm) | Unit weight (kN/m³) | Vertical spring coefficient of rail pad (kPa) |
|---|---|---|---|
| Rail pad | | | |
| | 5 | 11.5 | $15.3 \times 10^7$ |

## 3 Results and discussion

### 3.1 Roadbed settlement

The ground settlement in backfill area due to the excavation work has been estimated (Kojima et al., 2005; Kung et al., 2009; Ou et al., 2013) and its effect on responses of adjacent buildings has been investigated (Lin et al., 2017; Sabzi and Fakher, 2015; Schuster et al., 2009) have been widely studied. Clough and O'Rourke (1990) have proposed the method to estimate settlement in clay and sandy soils for in-situ wall systems using field measurement data and finite element analysis (Fig. 4). H, d, $\delta_{vm}$, and $\delta$ represent an excavation depth, a distance from the wall, the maximum settlement, and a settlement with respect to the distance, respectively. The settlements tend to average about 0.15% H. $\delta_{vm}$ occurs in the middle of excavation depth near the wall and a settlement linearly decreases as d increases. Little settlement occurs as d = 2H. Empirical correlations of settlement with d proposed by Bowels (1988) and Peck (1969) were similar to the one proposed by Clough and O'Rourke (1990). Bowels (1988) suggested that the settlements tend to average about 0.13 ~ 0.18% H. The magnitude of settlements is influenced by the ground stiffness, the wall stiffness, and support spacing. In this study, although ground is not fully excavated and also there are no wall systems, the settlement resulting from stress release in ground similarly occurs.

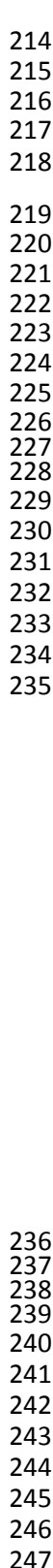

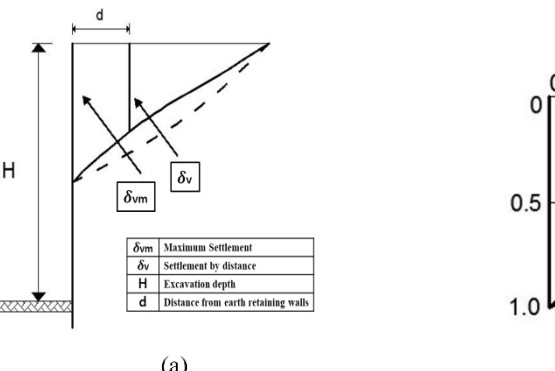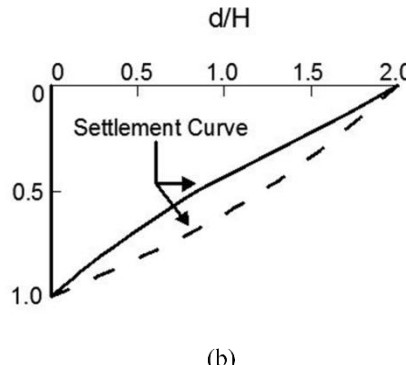

(a)                                                        (b)

**Figure 4.** Settlement of clay and sand backfill with respect to the distance from earth retaining walls: **(a)** Settlement of backfill and **(b)** Prediction of settlement

   The contours of ground settlement are presented for how the roadbed (Fig. 5) is influenced by a cavity adjacent to the urban railways. The contours of ground settlement are presented for cavities with diameters of 8 and 10 m, respectively, at a distance of 20 m between the center of the roadbed and the center of the cavity. As shown in the figures, ground settlement increases as the diameter of the cavity increases. As a cavity is generated on the right side of the roadbed, the right end of the roadbed is significantly settled down.

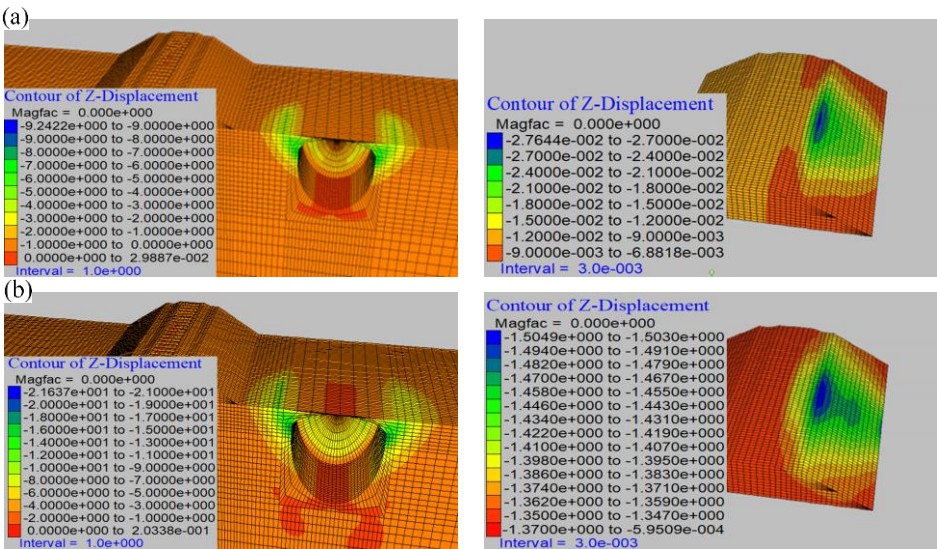

**Figure 5.** Vertical displacement contour of the roadbed at a distance of 20 m between the center of roadbed to the center of the cavity=20 m with respect to diameter of the cavity: **(a)** Diameter = 8 m and **(b)** Diameter = 10 m.

   The analysis results (Fig. 6) are presented for cavities with diameters of 4 -10 m. As the variation from 15 to 20 m in the distances between the center of the roadbed and the center of the cavity is applied to the 10 m cavity, roadbed settlements are calculated with respect to various diameters of the cavity. The cavity with a diameter of 10 m at a distance of 20 m has little influenced on the roadbed. However, as the diameter of the cavity at the same distance exceeds 10 m, the roadbed settlement exceeds the allowable value. As cavities with diameters of 8 and 6 m are generated, at distances less than 18 and 15 m, where d is close to or less than 2H (2D), it may exceed the allowable settlement resulting in an accident.
   Roadbed settlement increases as the diameter (D) of the cavity increases and the distance (d) between the roadbed and the cavity decreases. Therefore, in this study, the roadbed settlement is examined with respect to D normalized by d (Fig. 7). The regression analyses results show medium to high correlations of $r^2$=0.72. As D/d is greater than 0.2 and less than 0.3, the roadbed settlement is approximately 5 mm. It requires that a database of

measurement sensors should be established for real-time monitoring of the roadbed, structures and groundwater to
prevent disasters in advance. As D/d exceeds 0.35, the roadbed settlement substantially increases and is greater
than 10 mm. Since it may result in highly probable traffic accident, train operation should be stopped and the
roadbed should be reinforced or repaired.

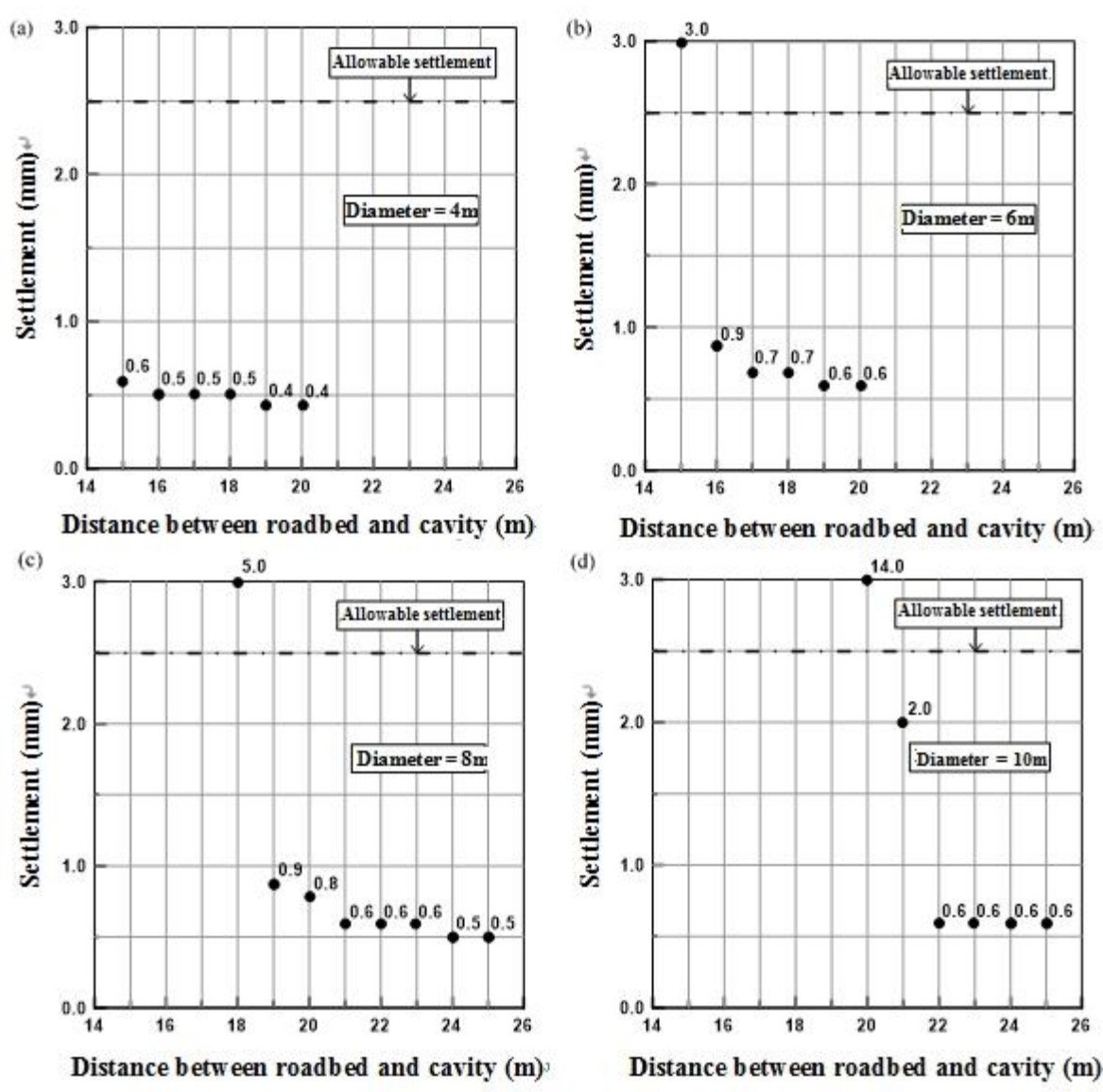

**Figure 6.** Roadbed settlement with respect to distance between roadbed and cavity: (**a**) Diameter = 4 m, (**b**) Diameter = 6 m,
(**c**) Diameter = 8 m, and (**d**) Diameter = 10 m.
The risk level has been estimated by the occurrence of roadbed settlements. Its risk level has been defined by the
value of the roadbed settlements relative to the allowable settlement. The risk level is defined as safe (not
problematic for both ride quality and track repair), caution (not problematic for track repair), warning (between
caution and danger), and danger (highly probable traffic accident) as a settlement is equal to or less than 2.5 mm,
greater than 2.5 mm and equal to or less than 4 mm, greater than 4 mm and equal to or less than 9mm, and greater
than 9 mm, respectively.

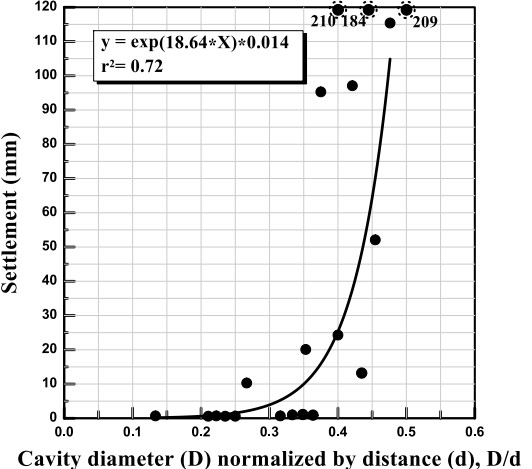

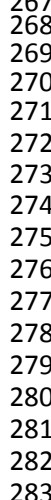

**Figure 7.** Regression analysis of roadbed settlements with respect to the diameter of the cavity and distance between roadbed
and the cavity


**3.2 Effects of groundwater level**

In this study, the effects of groundwater level on the roadbed settlement are examined and it is lowered until the
allowable settlement value of the roadbed is satisfied. The maximum distance between the roadbed and the cavity
for the analysis is determined as the maximum value for the satisfied allowable settlement with no groundwater
condition. A stability assessment of the roadbed has been carried out at the distance of 20 m for both 4 and 6 m
diameter cavities and at 25 m for both the 8 and 10 m diameter cavities.
The contours of ground settlement (Fig. 8) are presented to examine the groundwater level (GWL) effects in the
case of the 8 m diameter cavity located at a distance of 25 m from the roadbed to cavity. The contours of ground
settlement are presented with GWL on the ground surface and 20 m below it, respectively (Figs. 8a and 8b). The
settlement of the roadbed is highly subject to groundwater levels.

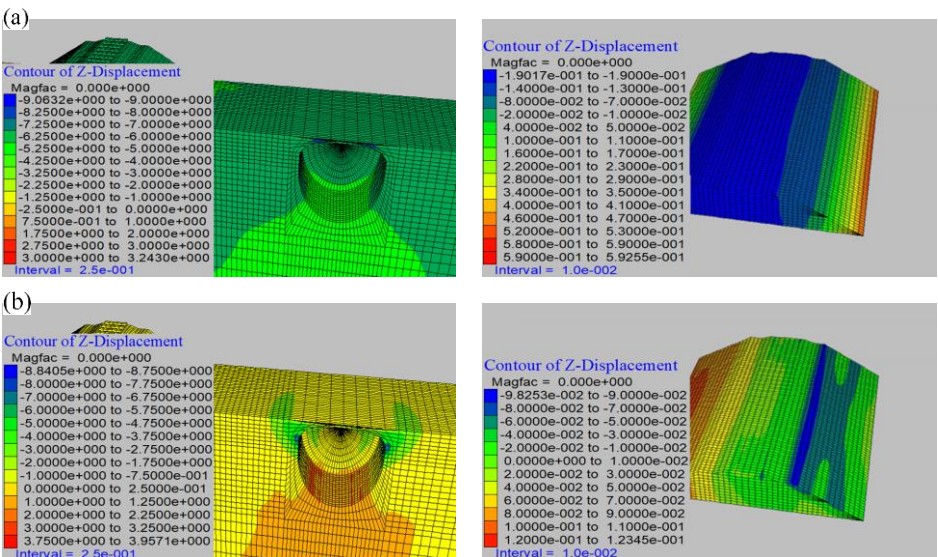

**Figure 8.** Vertical displacement contours of the roadbed for a cavity diameter 8 m, at the roadbed-to-cavity distance of 25 m: **(a)**
GWL = ground surface and **(b)** GWL = (-) 8 m.
The roadbed settlement (Fig. 9) is highly influenced by groundwater. Ground settlement for 4 and 6 m diameter
cavities located at a distance of 20 m from the roadbed (Figs. 9a and 9b) satisfies the allowable value for GWL = (-)
4 and (-) 12m, respectively. The ground settlement for 8 and 10 m diameter cavities located at a distance of 25 m
from the center of the roadbed (Figs. 9c and 9d) has substantially decreased as groundwater level is 8 and 15 m
below the ground surface, respectively, and satisfies the allowable value as its level is 18 and 22 m below the
ground surface, respectively. It indicates that a roadbed settlement is highly influenced by groundwater levels to an
extent greater than even the influence of the size of the cavity.

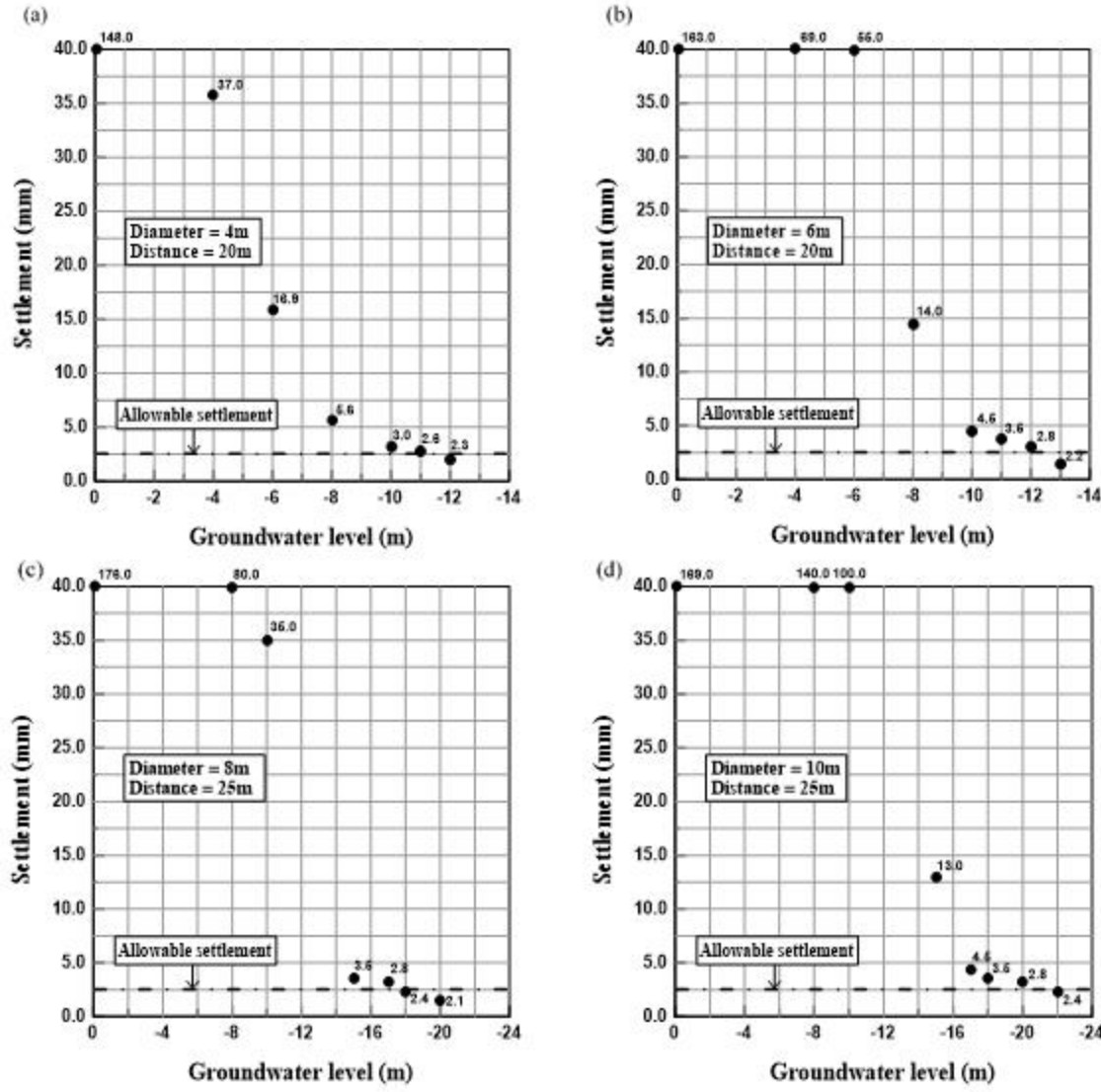

**Figure 9.** Roadbed settlement with respect to the groundwater level: **(a)** Diameter of the cavity = 4 m and distance of roadbed
from the center of the cavity = 20 m, **(b)** Diameter of the cavity = 6 m and distance of the roadbed from the center of cavity =20
m, **(c)** Diameter of the cavity = 8 m and distance of the roadbed from the center of cavity = 25 m, and **(d)** Diameter of the cavity
= 10 m and the distance of the roadbed from the center of the cavity = 25 m.

**3.3  Risk level assessment of roadbed**
Roadbed settlements induced by the cavity near urban railways have been estimated with respect to the

groundwater level, distance between the roadbed and cavity, and size of the cavity. As listed in Table 2, the roadbed settlement increases as the size of the cavity increases and the cavity is located close to the roadbed. As listed in Table 2, the roadbed settlement for no groundwater condition is less than the allowable value, whereas it is in extreme danger when groundwater is present. When it is in the status of danger, train operation should be stopped and the roadbed should be reinforced or repaired. When it is in the status of caution or warning, a database of measurement sensors for urban railways should be established for real-time monitoring of the roadbed, structures and groundwater for disaster prevention.

**Table 2.** Risk level of the roadbed with respect to the diameter of the cavity and the distance between the roadbed to the cavity for the groundwater condition

(Computation time: 3 weeks)

| | | Distance (m) | | | | | | | | | | |
|---|---|---|---|---|---|---|---|---|---|---|---|---|
| | | 25 | 24 | 23 | 22 | 21 | 20 | 19 | 18 | 17 | 16 | 15 |
| Diameter (m) | 10 | Case A | | Random Sampling | | | Danger | | | | | |
| | 8 | | | | | | | | | Danger | | |
| | 6 | | | Safety | | | | | | | | Danger |
| | 4 | | | | | | Case B | | | | | |

(Computation time: 4 days)

| | | | Groundwater Level (m) | | | | | | | |
|---|---|---|---|---|---|---|---|---|---|---|
| | | | -22 | -20 | -18 | -17 | -15 | -10 | -8 | Ground surface |
| Case A | Diameter of cavity (m) | 10 | | Caution | | Warning | | | | |
| | | 8 | Safety | | | | | | Danger | |

(Computation time: 3 days)

| | | | Groundwater Level (m) | | | | | | | |
|---|---|---|---|---|---|---|---|---|---|---|
| | | | -13 | -12 | -11 | -10 | -8 | -6 | -4 | Ground surface |
| Case B | Diameter of cavity (m) | 6 | | | | | | | | |
| | | 4 | Safety | | Caution | | Warning | | Danger | |

※ Safety(Settlement≤2.5mm), Caution(2.5mm<Settlement≤4.0mm), Warning(4.0mm<Settlement≤9.0mm), Danger(9.0mm<Settlement)

## 4 Conclusions

The number of occurrences of ground subsidence induced by a leakage of aged pipelines for water and sewage in urban areas resulting in various sizes of cavity near the urban railway in Seoul City has been found to increase and it may cause the roadbed settlement to exceed the allowable value. A large-scale cavity is rarely found, but if it is close to the roadbed, the roadbed is highly influenced by the cavity and may cause train derailment.

In this study, numerical analyses are carried out to estimate roadbed stability and its risk level associated with various groundwater levels, sizes of cavities. The analyses results show that roadbed settlement increases as the diameter (D) of the cavity increases and the distance (d) between the roadbed and the cavity decreases. The regression analyses results show that, as D/d is greater than 0.2 and less than 0.3, a database of measurement sensors should be established for real-time monitoring of the roadbed, structures and groundwater to prevent disasters in advance. As D/d exceeds 0.35, the roadbed settlement, which substantially increases and is in the status

of danger, may result in highly probable traffic accident. Therefore, train operation should be stopped and the
roadbed should be reinforced or repaired. The effects of groundwater level on the roadbed settlement are examined
at the distance of 20 m for both 4 and 6 m diameter cavities and at 25 m for both 8 and 10 m diameter cavities.
Ground settlement for 4 and 6 m diameter cavities located at a distance of 20 m from the roadbed satisfies the
allowable value for GWL = (-) 4 and (-) 12m, respectively. The ground settlement for 8 and 10 m diameter cavities
located at a distance of 25 m from the center of the roadbed has substantially decreased as GWL is 8 and 15 m
below the ground surface, respectively, and satisfies the allowable value as its level is 18 and 22 m below the
ground surface, respectively. It indicates that a roadbed settlement is highly influenced by groundwater levels to an
extent greater than even the influence of the size of the cavity.
*Acknowledgements.* This work was supported by the 2017 INJE University research grant.

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
