# Peer review of "Stability assessment of roadbed affected by ground subsidence adjacent to urban railways"

_Natural Hazards and Earth System Sciences, 2017_

## Referee Comment (RC1) · Anonymous Referee #1 · 19 Jan 2018

This study uses FLAC3D to assess the stability and risk level for the roadbed adjacent to urban railways with respect to various groundwater levels and the geometric characteristics of cavities. The topic is interesting and valuable. The manuscript may be potentially a good contribution for publication in Natural Hazards and Earth System Sciences after major revisions are addressed. Some questions are listed as follows:

(1) Abstract. Too many background descriptions are presented and too few results are found in the abstract. Four sentences from lines 13 to 17 in page 1 may be merged in a sentence. However some valuable results and conclusions, e.g., the road settlement and the effects of groundwater level, may be added.

(2) 1 Introduction.

[Figure]

- Discussion in the segment is not clear, and I think the segment is needed to be rewritten.

- A brief review is anticipated for the development of the software assessing the road risks. Discussions in the 2nd and 3rd paragraphs are chaotic.

- Figure 1 may be erased.

(3) 2 Case studies of ground subsidence. I think this is only an introduction of ground subsidence instead of the case studies of risk assessments. Hence the segment may be simplified and merged into the first segment Introduction. Moreover. Figures 2 and 3 may be merged.

(4) 3 Numerical analysis

- The principle of FLAC3D should be briefly and clearly described, or I can not believe what you calculated are reliable.

- Figure 4 is not clear especially as it is printed. Figure 5 may be erased for a similar description has been given in Figure 7.

- Might you try to simply tables 1-4 and merge them as a table? We might pay more attentions on results and discussion.

(5) 4 Roadbed Settlement and Stability

- Texts in Figure 7 are too small and blur.

- It's better that the number values of the vertical coordinates in Figures 8, 9, and11 grow from the bottom up. The unit of the horizontal ordinate may be added Figures 7-8.

- Line width of Figure 9 is different to others. What's the meaning of the horizontal ordinate caption in Figure 9?

- Lines 225-227 in page 9: Why could you define the risk level mentioned here?
- Tables 8-10: Color blocks in the tables are not clear as they are printed in black and white.

- In page 7, the heading of 4.1.1 should be erased.

- From segments 4.1 to 4.2, essential discussion on the problems related to the observed data may be added, and comparison of the results calculated in this study to other references may be replenished.

(6) 5 Conclusions. I could not find any quantitative conclusion here.

(7) The manuscript is readable, but still many minor language errors exist. For examples:

- In line 180, page 7, the original sentences are: "Diameter = 4m (a). Diameter = 6 m (b). Diameter = 8 m (c). Diameter = 10 m (d)". I think to merge the sentences as follows is better: "(a) Diameter = 4m, (b) Diameter = 6 m , (c) Diameter = 8 m, and (d) Diameter = 10 m"

- In line 207, page 8, "4-m and 6-m" may be revised as "4 and 6 m". ïĆǧ The sentence in the lines 253-255, page 10, is too complicated to understand.

---

## Referee Comment (RC2) · Anonymous Referee #2 · 25 Jan 2018

General comments This paper proposes an assessment on the stability and risk of the roadbed adjacent to urban railways by using a three-dimensional model. This topic could be interesting to NHESS readers, if some issues are more clearly addressed, in particular literatures review, research method, model's verification and discussions. For this reason, major revision of this paper is necessary before it can be considered for publication.

Specific comments P.1, 1 Introduction, please provide more literatures related to the assessment methods used in this topic, in particular the numerical models. P.2, 2 Case studies of ground subsidence, what kind of the cases are the simulated target in this paper? P.3, 3 Numerical analysis, please add a section to briefly introduce this three-dimensional numerical model such as theory base, essential parameters,

input/output, boundary conditions, initial conditions, etc. P.3, 3 Numerical analysis, please add a section of model's verification by historical events to properly demonstrate the reliability of the model's performance. P.3, 3.1 Conditions for numerical analysis, Ln.103-104, how to decide the scenario such as diameter 4-10 m, distance 15-25 m and various groundwater levels? Based on any field cases? In additions, please add a table to list total computational runs. P.3, Figure 4, what is the meaning of the roller attached on the left side and two sides of bottom? P.3, Figure 5, the legend texts are too small and unclear. Is it possible to merge this figure with Figure 4 as a single figure? P.4, 3.1.2 Physical properties of rail, rail pad, and prestressed concrete (PC) sleeper, too many tables in this section, I suggest to reorganize these tables to reduce table numbers. P.5, Figure 7 - The legend texts are too small and unclear. - Please use the same color interval of vertical displacement value of (a) and (b) in order to clearly to show "ground settlement increases as the diameter of the cavity increases". - Please keep the same geometric scale and view angle of the model display. P.6, 4.1.1 Regression analysis of roadbed settlement, too short descriptions. What's the meaning of the regression analysis? Why the groundwater level is absent in the regression? The better description for R-squared=0.72 probably is "medium to high correlation" instead of "high correlation". P.7, Figure 9, a linear equation in legend, editing error? P.7, Figure 10 - The legend texts are too small and unclear. - Why the vertical displacement is symmetry along the centerline of roadbed since only cavity on one side. P.9, It's difficult to understand the risk level through Table 5 – Table 7 since the risk level is based on the combination of cavity diameter, distance and groundwater level. I suggest to reorganize these table to perform more systematical outcome. P.10, 5 Conclusions, conclusions should include vital or quantitative findings of this paper.

---

## Author Comment (AC1) · 8 Mar 2018

The comment was uploaded in the form of a supplement:
https://www.nat-hazards-earth-syst-sci-discuss.net/nhess-2017-412/nhess-2017-412-AC1-supplement.pdf

**Referee #1**

1. **Abstract: Too many background descriptions are presented and too few results are found in the abstract. Four sentences from lines 13 to 17 in page 1 may be merged in a sentence. However some valuable results and conclusions, e.g., the roadbed settlement and the effects of groundwater level, may be added.**

   **P.1. Line 13-26:** Background descriptions are briefly presented and four sentences from lines 13 to 17 in page 1 are merged in a sentence. The roadbed settlement and the effects of groundwater level are added in Abstract.

   *In recent years, leakages in aged pipelines for water and sewage in urban areas have frequently induced ground loss resulting in cavities and ground subsidence causes the roadbed settlement greater than the allowable value. In this study, $FLAC^{3D}$, which is a three-dimensional finite-difference numerical modeling software, is used to do stability and risk level assessment for the roadbed in adjacent to urban railways with respect to various groundwater levels and the geometric characteristics of cavities. Numerical results show that roadbed settlement increases as the diameter (D) of the cavity increases and the distance (d) between the roadbed and the cavity decreases. The regression analyses results show that, as D/d is greater than 0.2 and less than 0.3, the roadbed is in the status of caution or warning. It requires a database of measurement sensors for real-time monitoring of the roadbed, structures and groundwater to prevent disasters in advance. As D/d exceeds 0.35, the roadbed settlement, which substantially increases and the roadbed is in the status of danger. Since it may result in highly probable traffic accident, train operation should be stopped and the roadbed should be reinforced or repaired. The effects of groundwater level on the roadbed settlement are examined and the analyses results indicate that a roadbed settlement is highly influenced by groundwater levels to an extent greater than even the influence of the size of the cavity.*

2. **Discussion in the segment is not clear, and I think the segment is needed to be rewritten**

   **P.11. Line 358-376:** Discussion is rewritten in detail.

[revised manuscript text omitted]

---

## Author Comment (AC2) · 8 Mar 2018

The comment was uploaded in the form of a supplement:
https://www.nat-hazards-earth-syst-sci-discuss.net/nhess-2017-412/nhess-2017-412-AC2-supplement.pdf
* * *
[Figure]

**Referee #2**

1. **P.1. 1.Introduction: more literature for assessment methods (numerical models)**

   **P.1. Line 35-50.:** Literature review for assessment methods (numerical models) is added

   *Risk management associated with safety is a fundamental focus in railway operations. It has been integrated into global safety management system of railways (Berrado et al., 2010) and developed to allow a rapid risk assessment using a common risk score matrix (Braband, 2011). As roadbed settlements exceed the allowable limits, it may result in track irregularity and derailments of trains causing heavy loss of life. Therefore, risk management tools are developed to deal with track safety by controlling and reducing the risk of derailments (Zarembski et al., 2006). In this study, methods to secure the stability of roadbeds have been examined using numerical analysis.*
   *Numerical analyses have been widely used for risk assessment. Numerical analyses using three-dimensional geotechnical codes were carried out to predict the subsidence area and its interaction with buildings (Castellanza et al., 2015) and a three-dimensional groundwater flow model for risk evaluation was developed to be an effective management strategy (Ashfaque et al., 2017). The coupling of numerical models and monitoring data contribute to undertake efficient risk reduction policies (Bozzano et al., 2013). Especially using FLAC, which is a finite-difference numerical code especially specialized in the area of geotechnical engineering, numerical computations to simulate the influence of rainfall (Pisani, 2010), both acoustic emission (AE) activities at AE sensor locations of the Kannagawa cavern (Cai et al., 2007), and a comprehensive pump test at Sellafield (Hakami, 2001) showed good agreement with field monitoring results. In this study, FLAC$^{3D}$, which is a three-dimensional finite-difference numerical code especially specialized in the area of geotechnical engineering, is adopted for numerical analysis.*

2. **P.2. 2. Case studies of ground subsidence, what kind of the cases are the simulated target in this paper?**

   **P.2. Line 62-70.:** The cases of ground subsidence occurred at nearby urban railways in South Korea are quite similar. Therefore, no specific case is selected for numerical analysis but the simulated cases cover historical events.

3. **P.3. 3. Numerical analysis, please add a section to briefly introduce this three-dimensional model such as theory base, essential parameters, input/output, boundary conditions, initial conditions, etc.**

   **P.3. Line 97-189:** FLAC3D is briefly introduced.

[revised manuscript text omitted]

---

## Editor Comment (EC1) · P. Tarolli (Editor) · 10 Mar 2018

Dear authors,

I strongly recommend to provide a suitable public reply to the reviewers' comments. It seems that you uploaded just the paper, not a reply to each comment raised. I will be able to make a decision, only after a clear public discussion.

Best regards

Paolo Tarolli

---

## Author Comment (AC3) · 11 Mar 2018

**Referee #1**

1. **Abstract: Too many background descriptions are presented and too few results are found in the abstract. Four sentences from lines 13 to 17 in page 1 may be merged in a sentence. However some valuable results and conclusions, e.g., the roadbed settlement and the effects of groundwater level, may be added.**

   **P.1. Line 13-26:** Background descriptions are briefly presented and four sentences from lines 13 to 17 in page 1 are merged in a sentence. The roadbed settlement and the effects of groundwater level are added in Abstract.

*In recent years, leakages in aged pipelines for water and sewage in urban areas have frequently induced ground loss resulting in cavities and ground subsidence causes the roadbed settlement greater than the allowable value. In this study, FLAC³ᴰ, which is a three-dimensional finite-difference numerical modeling software, is used to do stability and risk level assessment for the roadbed in adjacent to urban railways with respect to various groundwater levels and the geometric characteristics of cavities. Numerical results show that roadbed settlement increases as the diameter (D) of the cavity increases and the distance (d) between the roadbed and the cavity decreases. The regression analyses results show that, as D/d is greater than 0.2 and less than 0.3, the roadbed is in the status of caution or warning. It requires a database of measurement sensors for real-time monitoring of the roadbed, structures and groundwater to prevent disasters in advance. As D/d exceeds 0.35, the roadbed settlement, which substantially increases and the roadbed is in the status of danger. Since it may result in highly probable traffic accident, train operation should be stopped and the roadbed should be reinforced or repaired. The effects of groundwater level on the roadbed settlement are examined and the analyses results indicate that a roadbed settlement is highly influenced by groundwater levels to an extent greater than even the influence of the size of the cavity.*

2. **Discussion in the segment is not clear, and I think the segment is needed to be rewritten**

   **P.11. Line 358-376:** Discussion is rewritten in detail.

*The number of occurrences of ground subsidence induced by a leakage of aged pipelines for water and sewage in urban areas resulting in various sizes of cavity near the urban railway in Seoul City has been found to increase and it may cause the roadbed settlement to exceed the allowable value. A large-scale cavity is rarely found, but if it is close to the roadbed, the roadbed is highly influenced by the cavity and may cause train derailment.*

   *In this study, numerical analyses are carried out to estimate roadbed stability and its risk level associated with various groundwater levels, sizes of cavities. The analyses results show that roadbed settlement increases as the diameter (D) of the cavity increases and the distance (d) between the roadbed and the cavity decreases. The regression analyses results show that, as D/d is greater than 0.2 and less than 0.3, a database of measurement sensors should be established for real-time monitoring of the roadbed, structures and groundwater to prevent disasters in advance. As D/d exceeds 0.35, the roadbed settlement, which substantially increases and is in the status of danger, may result in highly probable traffic accident. Therefore, train operation should be stopped and the roadbed should be reinforced or repaired. The effects of groundwater level on the roadbed settlement are examined at the distance of 20 m for both 4 and 6 m diameter cavities and at 25 m for both 8 and 10 m diameter cavities. Ground settlement for 4 and 6 m diameter cavities located at a distance of 20 m from the roadbed satisfies the allowable value for GWL = (-) 4 and (-) 12m, respectively. The ground settlement for 8 and 10 m diameter cavities located at a distance of 25 m from the center of the roadbed has substantially decreased as GWL is 8 and 15 m below the ground surface, respectively, and satisfies the allowable value as its level is 18 and 22 m below the ground surface, respectively. It indicates that a roadbed settlement is highly influenced by groundwater levels to*

*an extent greater than even the influence of the size of the cavity.*

**3. A brief review is anticipated for the development of the software assessing the road risks. Discussion in the 2nd and 3rd paragraphs are chaotic. Figure 1 may be erased.**

**P.1. Line 35-50:** Literature review for risk assessment and numerical modeling is added.

*Risk management associated with safety is a fundamental focus in railway operations. It has been integrated into global safety management system of railways (Berrado et al., 2010) and developed to allow a rapid risk assessment using a common risk score matrix (Braband, 2011). As roadbed settlements exceed the allowable limits, it may result in track irregularity and derailments of trains causing heavy loss of life. Therefore, risk management tools are developed to deal with track safety by controlling and reducing the risk of derailments (Zarembski et al., 2006). In this study, methods to secure the stability of roadbeds have been examined using numerical analysis.*

*Numerical analyses have been widely used for risk assessment. Numerical analyses using three-dimensional geotechnical codes were carried out to predict the subsidence area and its interaction with buildings (Castellanza et al., 2015) and a three-dimensional groundwater flow model for risk evaluation was developed to be an effective management strategy (Ashfaque et al., 2017). The coupling of numerical models and monitoring data contribute to undertake efficient risk reduction policies (Bozzano et al., 2013). Especially using FLAC, which is a finite-difference numerical code especially specialized in the area of geotechnical engineering, numerical computations to simulate the influence of rainfall (Pisani, 2010), both acoustic emission (AE) activities at AE sensor locations of the Kannagawa cavern (Cai et al., 2007), and a comprehensive pump test at Sellafield (Hakami, 2001) showed good agreement with field monitoring results. In this study, $FLAC^{3D}$, which is a three-dimensional finite-difference numerical code especially specialized in the area of geotechnical engineering, is adopted for numerical analysis.*

**P.11. Line 358-376:** Discussion is rewritten in detail.

*The number of occurrences of ground subsidence induced by a leakage of aged pipelines for water and sewage in urban areas resulting in various sizes of cavity near the urban railway in Seoul City has been found to increase and it may cause the roadbed settlement to exceed the allowable value. A large-scale cavity is rarely found, but if it is close to the roadbed, the roadbed is highly influenced by the cavity and may cause train derailment.*

*In this study, numerical analyses are carried out to estimate roadbed stability and its risk level associated with various groundwater levels, sizes of cavities. The analyses results show that roadbed settlement increases as the diameter (D) of the cavity increases and the distance (d) between the roadbed and the cavity decreases. The regression analyses results show that, as D/d is greater than 0.2 and less than 0.3, a database of measurement sensors should be established for real-time monitoring of the roadbed, structures and groundwater to prevent disasters in advance. As D/d exceeds 0.35, the roadbed settlement, which substantially increases and is in the status of danger, may result in highly probable traffic accident. Therefore, train operation should be stopped and the roadbed should be reinforced or repaired. The effects of groundwater level on the roadbed settlement are examined at the distance of 20 m for both 4 and 6 m diameter cavities and at 25 m for both 8 and 10 m diameter cavities. Ground settlement for 4 and 6 m diameter cavities located at a distance of 20 m from the roadbed satisfies the allowable value for GWL = (-) 4 and (-) 12m, respectively. The ground settlement for 8 and 10 m diameter cavities located at a distance of 25 m from the center of the roadbed has substantially decreased as GWL is 8 and 15 m below the ground surface, respectively, and satisfies the allowable value as its level is 18 and 22 m below the ground surface, respectively. It indicates that a roadbed settlement is highly influenced by groundwater levels to an extent greater than even the influence of the size of the cavity.*

**P.1:** Figure 1 is erased.

**4. 2 Case studies of ground subsidence. I think this is only an introduction of ground subsidence instead of the case studies of risk assessments. Hence the segment may be simplified and merged into the first segment introduction.**

**P.1.Line 30-95:** The segment is simplified and merged into the first segment introduction.

**5. Moreover, Figures 2 and 3 may be merged**

**P.2.:** Figures 2 and 3 are merged to Figure 1.

**6. The principle of FLAC3D should be briefly and clearly described, or I cannot believe what you calculated are reliable**

**P.3. Line 97-189:** The principle of FLAC3D is described.

**2 Numerical analysis**

[revised manuscript text omitted]

*in which $d[\sigma]/dt$ is the material time derivative of [$\sigma$], and [ω] is the rate of rotation tensor.*

**7.   Figure 4 is not clear especially as it is printed.**

**P.5.:** Figure 2 is magnified to be clearly presented.

**8.   Figure 5 may be erased for a similar description has been given in Figure 7**

**P.5.:** Figure 5 is erased.

**9.   Might you try to simply tables 1-4 and merge them as a table?**

**P.6.:** Tables 1-4 is merged to Table 1.

**10.  We might pay more attentions on results and discussion**

**P.7. Line 280-287:** Results are rewritten in detail.

*Roadbed settlement increases as the diameter (D) of the cavity increases and the distance (d) between the roadbed and the cavity decreases. Therefore, in this study, the roadbed settlement is examined with respect to D normalized by d (Fig. 7). The regression analyses results show medium to high correlations of $r^2=0.72$. As D/d is greater than 0.2 and less than 0.3, the roadbed settlement is approximately 5 mm. It requires that a database of measurement sensors should be established for real-time monitoring of the roadbed, structures and groundwater to prevent disasters in advance. As D/d exceeds 0.35, the roadbed settlement substantially increases and is greater than 10 mm. Since it may result in highly probable traffic accident, train operation should be stopped and the roadbed should be reinforced or repaired.*

**P.11. Line 358-376:** Discussion is rewritten in detail.

*The number of occurrences of ground subsidence induced by a leakage of aged pipelines for water and sewage in urban areas resulting in various sizes of cavity near the urban railway in Seoul City has been found to increase and it may cause the roadbed settlement to exceed the allowable value. A large-scale cavity is rarely found, but if it is close to the roadbed, the roadbed is highly influenced by the cavity and may cause train derailment.*
*In this study, numerical analyses are carried out to estimate roadbed stability and its risk level associated with various groundwater levels, sizes of cavities. The analyses results show that roadbed settlement increases as the diameter (D) of the cavity increases and the distance (d) between the roadbed and the cavity decreases. The regression analyses results show that, as D/d is greater than 0.2 and less than 0.3, a database of measurement sensors should be established for real-time monitoring of the roadbed, structures and groundwater to prevent disasters in advance. As D/d exceeds 0.35, the roadbed settlement, which substantially increases and is in the status of danger, may result in highly probable traffic accident. Therefore, train operation should be stopped and the roadbed should be reinforced or repaired. The effects of groundwater level on the roadbed settlement are examined at the distance of 20 m for both 4 and 6 m diameter cavities and at 25 m for both 8 and 10 m diameter cavities. Ground settlement for 4 and 6 m diameter cavities located at a distance of 20 m from the roadbed satisfies the allowable value for GWL = (-) 4 and (-) 12m,   respectively. The ground settlement for 8 and 10 m diameter cavities located at a distance of 25 m from the center of the roadbed has substantially decreased as GWL is 8 and 15 m below the ground surface, respectively, and satisfies the allowable value as its level is 18 and 22 m below the ground surface, respectively. It indicates that a roadbed settlement is highly influenced by groundwater levels to an extent greater than even the influence of the size of the cavity.*

**11. Texts in Figure 7 are too small and blur**

**P.7.:** Texts in Figure 7 are magnified to be clearly presented.

**12. It's better that the number values of the vertical coordinates in Figures 8, 9, and 11 grow from the bottom up.**

**P.8. & P.10.:** In general, settlement starts from the top as shown in Figures 5 & 8.

**P.9.:** However, as shown in Figure 6, an origin of the vertical coordinates is positioned at the bottom for the purpose of regression analysis.

13. **The unit of the horizontal ordinate may be added Figures 7-8.**

    **P.8.:** The unit is added in Figure 5.

14. **Line width of Figure 9 is different to others**

    **P.9.:** Line width of Figure 7 is changed to be consistent with others.

15. **What is the meaning of the horizontal ordinate caption in Figure 9**

    **P.9.:** In Figure 6, Caption in horizontal axis is added..

16. **Lines 225-227 in page 9: Why could you define the risk level mentioned here?**

    **P.7. Line 296-299:** Definition of the risk level is moved to section 3.2.

17. **Tables 8-10: Color blocks in the tables are not clear as they are printed in black and white**

    **P.11.:** Colors in Tables 8-10 are changed to black and white in Table 2.

18. **From Segments 4.1 to 4.2, essential discussion on the problems related to the observed data may be added, and comparison of the results calculated in this study to other references may be replenished.**

    **P.6-10.:** Unfortunately, any observed data obtained from other references for roadbed settlement associated with cavity have not been found.

19. **I could not find any quantitative conclusions here**

    **P.11. Line 358-376:** Conclusions are quantitatively described.

    *The number of occurrences of ground subsidence induced by a leakage of aged pipelines for water and sewage in urban areas resulting in various sizes of cavity near the urban railway in Seoul City has been found to increase and it may cause the roadbed settlement to exceed the allowable value. A large-scale cavity is rarely found, but if it is close to the roadbed, the roadbed is highly influenced by the cavity and may cause train derailment.*
    *In this study, numerical analyses are carried out to estimate roadbed stability and its risk level associated with various groundwater levels, sizes of cavities. The analyses results show that roadbed settlement increases as the diameter (D) of the cavity increases and the distance (d) between the roadbed and the cavity decreases. The regression analyses results show that, as D/d is greater than 0.2 and less than 0.3, a database of measurement sensors should be established for real-time monitoring of the roadbed, structures and groundwater to prevent disasters in advance. As D/d exceeds 0.35, the roadbed settlement, which substantially increases and is in the status of danger, may result in highly probable traffic accident. Therefore, train operation should be stopped and the roadbed should be reinforced or repaired. The effects of groundwater level on the roadbed settlement are examined at the distance of 20 m for both 4 and 6 m diameter cavities and at 25 m for both 8 and 10 m diameter cavities. Ground settlement for 4 and 6 m diameter cavities located at a distance of 20 m from the roadbed satisfies the allowable value for GWL = (-)*

*4 and (-) 12m, respectively. The ground settlement for 8 and 10 m diameter cavities located at a distance of 25 m from the center of the roadbed has substantially decreased as GWL is 8 and 15 m below the ground surface, respectively, and satisfies the allowable value as its level is 18 and 22 m below the ground surface, respectively. It indicates that a roadbed settlement is highly influenced by groundwater levels to an extent greater than even the influence of the size of the cavity.*

20. **The manuscript is readable, but still many minor language errors exist. For examples: In line 180, page 7, the original sentences are: "Diameter = 4m (a). Diameter = 6 m (b). Diametr = 8 m (c). Diameter = 10 m (d)" I think to merge the sentences as follows is better: "(a) Diameter = 4m, (b) Diameter = 6 m, (c) Diamter = 8 m, and (d) Diameter = 10 m"**

**P.8. Line 301-302:** Captions in Figure 6 are changed.

*Figure 5. Roadbed settlement with respect to distance between roadbed and cavity: (a) Diameter = 4 m, (b) Diameter = 6 m, (c) Diameter = 8 m, and (d) Diameter = 10 m.*

21. **In line 207, page 8: "4-m and 6-m" may be revised as "4 and 6 m.**

**P.8. Line 304:** "4-m and 6-m" is changed to "4 and 6 m. Errors similar to this are corrected.

22. **The sentence in the lines 253-255, page 10, is too complicated to understand**

**P.11. Line 358-376:** Conclusions are quantitatively described.

*The number of occurrences of ground subsidence induced by a leakage of aged pipelines for water and sewage in urban areas resulting in various sizes of cavity near the urban railway in Seoul City has been found to increase and it may cause the roadbed settlement to exceed the allowable value. A large-scale cavity is rarely found, but if it is close to the roadbed, the roadbed is highly influenced by the cavity and may cause train derailment.*

*In this study, numerical analyses are carried out to estimate roadbed stability and its risk level associated with various groundwater levels, sizes of cavities. The analyses results show that roadbed settlement increases as the diameter (D) of the cavity increases and the distance (d) between the roadbed and the cavity decreases. The regression analyses results show that, as D/d is greater than 0.2 and less than 0.3, a database of measurement sensors should be established for real-time monitoring of the roadbed, structures and groundwater to prevent disasters in advance. As D/d exceeds 0.35, the roadbed settlement, which substantially increases and is in the status of danger, may result in highly probable traffic accident. Therefore, train operation should be stopped and the roadbed should be reinforced or repaired. The effects of groundwater level on the roadbed settlement are examined at the distance of 20 m for both 4 and 6 m diameter cavities and at 25 m for both 8 and 10 m diameter cavities. Ground settlement for 4 and 6 m diameter cavities located at a distance of 20 m from the roadbed satisfies the allowable value for GWL = (-) 4 and (-) 12m, respectively. The ground settlement for 8 and 10 m diameter cavities located at a distance of 25 m from the center of the roadbed has substantially decreased as GWL is 8 and 15 m below the ground surface, respectively, and satisfies the allowable value as its level is 18 and 22 m below the ground surface, respectively. It indicates that a roadbed settlement is highly influenced by groundwater levels to an extent greater than even the influence of the size of the cavity.*

---

## Author Comment (AC5) · 11 Mar 2018

**Referee #2**

**1. P.1. 1.Introduction: more literature for assessment methods (numerical models)**

**P.1. Line 35-50.:** Literature review for assessment methods (numerical models) is added

*Risk management associated with safety is a fundamental focus in railway operations. It has been integrated into global safety management system of railways (Berrado et al., 2010) and developed to allow a rapid risk assessment using a common risk score matrix (Braband, 2011). As roadbed settlements exceed the allowable limits, it may result in track irregularity and derailments of trains causing heavy loss of life. Therefore, risk management tools are developed to deal with track safety by controlling and reducing the risk of derailments (Zarembski et al., 2006). In this study, methods to secure the stability of roadbeds have been examined using numerical analysis.*

*Numerical analyses have been widely used for risk assessment. Numerical analyses using three-dimensional geotechnical codes were carried out to predict the subsidence area and its interaction with buildings (Castellanza et al., 2015) and a three-dimensional groundwater flow model for risk evaluation was developed to be an effective management strategy (Ashfaque et al., 2017). The coupling of numerical models and monitoring data contribute to undertake efficient risk reduction policies (Bozzano et al., 2013). Especially using FLAC, which is a finite-difference numerical code especially specialized in the area of geotechnical engineering, numerical computations to simulate the influence of rainfall (Pisani, 2010), both acoustic emission (AE) activities at AE sensor locations of the Kannagawa cavern (Cai et al., 2007), and a comprehensive pump test at Sellafield (Hakami, 2001) showed good agreement with field monitoring results. In this study, $FLAC^{3D}$, which is a three-dimensional finite-difference numerical code especially specialized in the area of geotechnical engineering, is adopted for numerical analysis.*

**2. P.2. 2. Case studies of ground subsidence, what kind of the cases are the simulated target in this paper?**

**P.2. Line 62-70.:** The cases of ground subsidence occurred at nearby urban railways in South Korea are quite similar. Therefore, no specific case is selected for numerical analysis but the simulated cases cover historical events.

**3. P.3. 3. Numerical analysis, please add a section to briefly introduce this three-dimensional model such as theory base, essential parameters, input/output, boundary conditions, initial conditions, etc.**

**P.3. Line 97-189:** FLAC3D is briefly introduced.

**2 Numerical analysis**

[revised manuscript text omitted]

**4. P.3. 3. Numerical analysis, please add a section of model's verification by historical events to properly demonstrate the reliability of the model's performance**

**P.1. Line 41-50:** Model's verification by historical events are added to demonstrate the reliability of the model's performance.

*Numerical analyses have been widely used for risk assessment. Numerical analyses using three-dimensional geotechnical codes were carried out to predict the subsidence area and its interaction with buildings (Castellanza et al., 2015) and a three-dimensional groundwater flow model for risk evaluation was developed to be an effective management strategy (Ashfaque et al., 2017). The coupling of numerical models and monitoring data contribute to undertake efficient risk reduction policies (Bozzano et al., 2013). Especially using FLAC, which is a finite-difference numerical code especially specialized in the area of geotechnical engineering, numerical computations to simulate the influence of rainfall (Pisani, 2010), both acoustic emission (AE) activities at AE sensor locations of the Kannagawa cavern (Cai et al., 2007), and a comprehensive pump test at Sellafield (Hakami, 2001) showed good agreement with field monitoring results. In this study, $FLAC^{3D}$, which is a three-dimensional finite-difference numerical code especially specialized in the area of geotechnical engineering, is adopted for numerical analysis.*

**5. P.3, 3.1 Conditions for numerical analysis, Ln. 103-104, how to decide the scenario such as diameter 4-10 m, distance 15-25 m and various groundwater levels? Based on any field cases?**

**P.5. Line 194-197:** Diameters of the cavity are determined by historical events described in Introduction. Distance and groundwater level are arbitrarily determined with respect to diameter of the cavity.

*A circular cavity below the ground surface has been modeled with respect to diameters (D) of 4-10 m, which is selected by historical events as described in previous section. Distances of 15-25 m from the cavity to the center of the roadbed and various groundwater levels are arbitrarily selected for roadbed settlement influenced by given size of cavity.*

**6. P.3, In additions, please add a table to list total computational runs.**

**P.11.:** Total computation time is added in Table 2.

**7. P.3, Figure 4, what is the meaning of the roller attached on the left side and two sides of bottom?**

**P.5. Line 198-200:** The meaning of the roller is added.

*As shown in the figure, roller supports prevent normal translations, but capable of tangential translations and/or rotations. There is a single linear reaction force in either vertical or horizontal directions.*

**8. P.3, Figure 5, the legend texts are too small and unclear. Is it possible to merge this figure with Figure 4 as a single figure?**

**P.7.:** Figure 5 (original manuscript) is erased because a similar description is given in Figure 4 (P.7. revised manuscript).

**9. P.4, 3.1.2 Physical properties of rail, rail pad, and prestressed, too many tables in this section, I suggest to reorganize these tables to reduce table numbers**

**P.6.:** Tables 1-4 are reorganized and merged to Table 1.

**10. P.5, Figure 7 - The legend texts are too small and unclear. – Please use the same color interval of vertical displacement value of (a) and (b) in order to clearly to show "ground settlement increases as the diameter of the cavity increases". – Please keep the same geometric scale and view angle of the model display.**

**P.7.:** In Figure 4, the legend texts are changed to be clearly visible and the same color interval of vertical displacement and the same geometric scale and view angle of the model are used.

**11. P.6, 4.1.1 Regression analysis of roadbed settlement, too short descriptions. What's the meaning of the regression analysis? Why the groundwater level is absent in the regression?**

**P.7. Line 280-287:** Meaning of regression analysis is described in detail. In dry condition, regression analysis of roadbed settlement associated with distance and diameter is carried out. If the groundwater level is included for the analysis, there are too many parameters to define its relationships.

*Roadbed settlement increases as the diameter (D) of the cavity increases and the distance (d) between the roadbed and the cavity decreases. Therefore, in this study, the roadbed settlement is examined with respect to D normalized by d (Fig. 7). The regression analyses results show medium to high correlations of $r^2=0.72$. As D/d is greater than 0.2 and less than 0.3, the roadbed settlement is approximately 5 mm. It*

*requires that a database of measurement sensors should be established for real-time monitoring of the roadbed, structures and groundwater to prevent disasters in advance. As D/d exceeds 0.35, the roadbed settlement substantially increases and is greater than 10 mm. Since it may result in highly probable traffic accident, train operation should be stopped and the roadbed should be reinforced or repaired.*

**12. The better description for R-squared=0.72 probably is "medium to high correlation" instead of "high correlation".**

**P.7. Line 282:** It is changed to "medium to high correlation" instead of "high correlation".

**13. P.7, Figure 9, a linear equation in legend, editing error?**

**P.9.:** In Figure 6, It is corrected to exponential equation.

**14. P.7, Figure 10 – The legend texts are too small and unclear. – Why the vertical displacement is symmetry along the centerline of roadbed since only cavity on one side**

**P.9.:** Figures 4 & 7 are corrected and the right and the left side of roadbed settle down and heave, respectively, because the cavity is on the right side.

**15. P.9, It's difficult to understand the risk level through Table 5-Table 7 since the risk level is based on the combination of cavity diameter, distance and groundwater level. I suggest to reorganize these tables to perform more systematical outcome.**

**P.11.:** Tables 5-7 (original manuscript) are merged to Table 2 (revised manuscript) and reorganized in systematic pattern.

**16. P.10, 5 Conclusions, conclusions should include vital or quantitative findings of this paper.**

**P.11. Line 358-376:** Quantitatively findings are described in conclusions.

*The number of occurrences of ground subsidence induced by a leakage of aged pipelines for water and sewage in urban areas resulting in various sizes of cavity near the urban railway in Seoul City has been found to increase and it may cause the roadbed settlement to exceed the allowable value. A large-scale cavity is rarely found, but if it is close to the roadbed, the roadbed is highly influenced by the cavity and may cause train derailment.*

*In this study, numerical analyses are carried out to estimate roadbed stability and its risk level associated with various groundwater levels, sizes of cavities. The analyses results show that roadbed settlement increases as the diameter (D) of the cavity increases and the distance (d) between the roadbed and the cavity decreases. The regression analyses results show that, as D/d is greater than 0.2 and less than 0.3, a database of measurement sensors should be established for real-time monitoring of the roadbed, structures and groundwater to prevent disasters in advance. As D/d exceeds 0.35, the roadbed settlement, which substantially increases and is in the status of danger, may result in highly probable traffic accident. Therefore, train operation should be stopped and the roadbed should be reinforced or repaired. The effects of groundwater level on the roadbed settlement are examined at the distance of 20 m for both 4 and 6 m diameter cavities and at 25 m for both 8 and 10 m diameter cavities. Ground settlement for 4 and 6 m diameter cavities located at a distance of 20 m from the roadbed satisfies the allowable value for GWL = (-) 4 and (-) 12m, respectively. The ground settlement for 8 and 10 m diameter cavities located at a distance of 25 m from the center of the roadbed has substantially decreased as GWL is 8 and 15 m below the*

*ground surface, respectively, and satisfies the allowable value as its level is 18 and 22 m below the ground surface, respectively. It indicates that a roadbed settlement is highly influenced by groundwater levels to an extent greater than even the influence of the size of the cavity.*

---

## Author Response (AR2)

**Referee #1**

(1) "2. Numerical analysis". I think the section may be largely simplified. The level 2 subtitle "2.3 Allowable settlement of the roadbed" may be erased. All of the level 3 subtitle, e.g. 2.1.1, 2.2.2, may be erased.
Yes, I asked you to describe the principle of FLAC3D, but please note that the description should be briefly and clearly described. I think 1/3 page or so is enough which includes a text description on the principle of the FLAC3D and not more than three equations.
May "2.2 Conditions for numerical analysis" be simplified more? Is Figure 3 essential?

**P.3. Line 102-124:** All the level 3 subtitles are erased. The principle of FLAC3D is briefly and clearly described with 1/3 page and two equations. Section 2.2 describes conditions for numerical analysis clearly and Figure 3 represents an applied loading condition. If there is no harm to leave it, I may leave it as it is.

**2.1 Theoretical background of FLAC$^{3D}$**

FLAC$^{3D}$ (Fast Lagrangian Analysis of Continua in three Dimensions) is numerical modeling software for advanced geotechnical analysis of soil, rock, groundwater, and ground support in three dimensions. FLAC is used for analysis, testing, and design by geotechnical, civil, and mining engineers (Itasca Consulting Group Inc., 2002). It is designed to accommodate any kind of geotechnical engineering project that requires continuum analysis. The mechanics of the medium are derived from general principles (definition of strain, laws of motion), and the use of constitutive equations defining the idealized material. The resulting mathematical expression is a set of partial differential equations, relating mechanical (stress) and kinematic (strain rate, velocity) variables, which are to be solved for particular geometries and properties, given specific boundary and initial conditions. An important aspect of the model is the inclusion of the equations of motion, although FLAC3D is primarily concerned with the state of stress and deformation of the medium near the state of equilibrium. Application of the continuum form of the momentum principle yields Cauchy's equations of motion:

$$\sigma_{ij,j} + \rho b_i = \rho(d_{vi}/d_t) \tag{1}$$

Where $\sigma$ is the symmetric stress tensor, $\rho$ is the mass per unit volume of the medium, $[b]$ is the body force per unit mass, and $d[v]/dt$ is the material derivative of the velocity. These laws govern, in the mathematical model, the motion of an elementary volume of the medium from the forces applied to it. Note that in the case of static equilibrium of the medium, the acceleration $d[v]/dt$ is zero, and Eq. (1) reduces to the partial differential equations of equilibrium:

$$\sigma_{ij,j} + \rho b_i = 0 \tag{2}$$

(2) "3 Roadbed Settlement and Stability". May we change the title to " 3 Results and discussion"? The section may be greatly enriched. So many figures are given and too few discussions are found. Besides the results you have calculated, you could also discuss the REASONS why the results are right, and the EFFECTS on the real engineering of the calculated results. I am sorry I do not agree to your opinion that "any observed data obtained from other references for roadbed settlement associated with cavity have not been found". PLEASE READ MORE REFERENCES AND THEN GIVE A MORE IN-DEPTH DISCUSSION.

**P.5. Line 196:** Title is changed to "3 Results and discussion".

**P.5. Line 200-228 & P.6. Line 214:** Nine references are added and more depth discussions are carried out. Previous researches related to settlement adjacent to excavation work are described and Figure 4 is added.

*The ground settlement in backfill area due to the excavation work has been estimated (Kojima et al., 2005; Kung et al., 2009; Ou et al., 2013) and its effect on responses of adjacent buildings has been investigated (Lin et al., 2017; Sabzi and Fakher, 2015; Schuster et al., 2009). Clough and O'Rourke (1990) have proposed the method to estimate settlement in clay and sandy soils for in-situ wall systems using field measurement data and finite element analysis (Fig. 4). H, d, $\delta_{vm}$, and $\delta$ represent an excavation depth, a distance from the wall, the maximum settlement, and a settlement with respect to the distance, respectively. The settlements tend to average about 0.15% H. $\delta_{vm}$ occurs in the middle of excavation depth near the wall and a settlement linearly decreases as d increases. Little settlement occurs as d = 2H. Empirical correlations of settlement with d proposed by Bowels (1988) and Peck (1969) were similar to the one proposed by Clough and O'Rourke (1990). Bowels (1988) suggested that the settlements tend to average about 0.13 ~ 0.18% H. The magnitude of settlements is influenced by the ground stiffness, the wall stiffness, and support spacing. In this study, although ground is not fully excavated and also there are no wall systems, the settlement resulting from stress release in ground similarly occurs.*

**P.6. Line 244-246:** The REASONS why the results are right are explained.

*As cavities with diameters of 8 and 6 m are generated, at distances less than 18 and 15 m, where d is close to or less than 2H (2D), it may exceed the allowable settlement resulting in an accident.*

**P.6. Line 249-254:** The EFFECTS on the real engineering of the calculated results are discussed.

*As D/d is greater than 0.2 and less than 0.3, the roadbed settlement is approximately 5 mm. It requires that a database of measurement sensors should be established for real-time monitoring of the roadbed, structures and groundwater to prevent disasters in advance. As D/d exceeds 0.35, the roadbed settlement substantially increases and is greater than 10 mm. Since it may result in highly probable traffic accident, train operation should be stopped and the roadbed should be reinforced or repaired.*

(3) It's better that the number values of the vertical coordinates in Figures 5 and 8 grow from the bottom to the top. Figure 6 is good.

**P.7. Line 255 & P.9. Line 295:** The number values of the vertical coordinates in Figures 6 and 9 are changed to grow from the bottom to the top.

**Referee #2**

I suggest authors should try to perform more academic discussion in particular parameters calibration/sensitivity or providing the references of parameters calibrated in a similar geo-environment area.

**P.5. Line 200-228 & P.6. Line 214:** Nine references are added and academic discussion is carried out by providing the references of parameters calibrated in a similar geo-environment area. Previous researches related to settlement adjacent to excavation work are described and Figure 4 is added.

*The ground settlement in backfill area due to the excavation work has been estimated (Kojima et al., 2005; Kung et al., 2009; Ou et al., 2013) and its effect on responses of adjacent buildings has been investigated (Lin et al., 2017; Sabzi and Fakher, 2015; Schuster et al., 2009). Clough and O'Rourke (1990) have proposed the method to estimate settlement in clay and sandy soils for in-situ wall systems using field measurement data and finite element analysis (Fig. 4). H, d, $\delta_{vm}$, and $\delta$ represent an excavation depth, a distance from the wall, the maximum settlement, and a settlement with respect to the distance, respectively. The settlements tend to average about 0.15% H. $\delta_{vm}$ occurs in the middle of excavation depth near the wall and a settlement linearly decreases as d increases. Little settlement occurs as d = 2H. Empirical correlations of settlement with d proposed by Bowels (1988) and Peck (1969) were similar to the one proposed by Clough and O'Rourke (1990). Bowels (1988) suggested that the settlements tend to average about 0.13 ~ 0.18% H. The magnitude of settlements is influenced by the ground stiffness, the wall stiffness, and support spacing. In this study, although ground is not fully excavated and also there are no wall systems, the settlement resulting from stress release in ground similarly occurs.*

**P.6. Line 244-246:** The reasons why the results are right are explained.

*As cavities with diameters of 8 and 6 m are generated, at distances less than 18 and 15 m, where d is close to or less than 2H (2D), it may exceed the allowable settlement resulting in an accident.*

**P.6. Line 249-254:** The effects on the real engineering of the calculated results are discussed.

*As D/d is greater than 0.2 and less than 0.3, the roadbed settlement is approximately 5 mm. It requires that a database of measurement sensors should be established for real-time monitoring of the roadbed, structures and groundwater to prevent disasters in advance. As D/d exceeds 0.35, the roadbed settlement substantially increases and is greater than 10 mm. Since it may result in highly probable traffic accident, train operation should be stopped and the roadbed should be reinforced or repaired.*

---

## Author Response (AR3)

**Dr. Paolo Tarolli**

From my side I have an additional technical correction: an improvement of fig. 5 and fig. 8 is necessary. I would suggest making these clearer in details and dpi. Also I suggest, if it is possible, to post outside the figure the legends, avoiding the overlap with the background. Is it even possible to simplify the numbers of the legend with some decimal exponents?

(a) *Figures 5 and 8 have been improved and it is the best quality to be produced from FLAC3D.*

(b) *If the legend is outside of the figure, it looks scattered and the figure should be relatively smaller size because the legend should be large enough to be visualized. The legend doesn't block relatively important contour or blocks no change of contour, the overlap with the background is ignorable. Based on these, it is inside of the figure.*

(c) *The numbers are simplified.*

**Referee #1 (accepted subject to technical corrections)**

(1)   The manuscript, entitled, Dangerous degree forecast of soil and water loss on highway slopes in mountainous areas using RUSLE model, has been greatly improved. I think the paper could be accepted for publication if language errors have been revised.

Still there are many language errors throughout the paper. For example, In lines 97-99, page3, you said: "Chen (2010) according to terrain characteristics of roadbed side slope…"; In lines 106-109, page 3, you said: " Zhang (2016) investigated the spatio-temporal…he used land…was collected to derive". I am lost. In addition, in line 52, page 2: Figure 2 should emerge after Figure 1.

(1) *Comments mentioned above are related to someone else's article.*

**Referee #2 (accepted subject to technical corrections)**

(1)     Please improve the clarity of Fig.6 and 9.

*(1) Figures 6 and 9 are highly improved with the clarity.*